# Masters of Gene Expression: Transcription Factors in Pediatric Cancers

**DOI:** 10.3390/cancers17213439

**Published:** 2025-10-27

**Authors:** Anup S. Pathania

**Affiliations:** Department of Pharmacology and Experimental Neuroscience, University of Nebraska Medical Center, Omaha, NE 68198, USA; anup.pathania@unmc.edu

**Keywords:** pediatric cancer, transcription factors, chromosomal rearrangements, gene regulation, targeted therapy, PROTACs, molecular glue degraders, epigenetic regulation

## Abstract

Transcription factors (TFs) are proteins that directly regulate gene expression and are crucial for cell development and differentiation. When TFs become abnormally active, they can drive uncontrolled cancer growth and survival. Frequent disruption of TFs is a striking feature of pediatric cancers, despite their low overall mutation burden. These altered TFs interfere with normal lineage programs and “hijack” developmental signaling pathways, creating self-sustaining loops that promote tumor growth. Studying TFs is therefore essential. It helps identify biomarkers for diagnosis, predict treatment responses, and develop therapies that directly target TFs or their regulatory networks. These insights offer new opportunities to design more effective and less toxic treatments for pediatric patients.

## 1. Introduction

TFs are proteins that regulate gene activity by binding to specific DNA sequences within promoter and enhancer regions, thereby turning genes on or off. These binding sites, known as consensus sequences, are evolutionarily conserved and recognized based on both nucleotide composition and DNA structural features such as groove width, helical twist, and flexibility. Epigenetic modifications like DNA methylation and histone marks further fine-tune TF binding by modulating chromatin accessibility [1,2,3]. TFs orchestrate cell type specific gene expression programs essential for cellular identity, differentiation, and function. When dysregulated, they can act as oncogenes or tumor suppressors, influencing virtually every hallmark of cancer, including proliferation, apoptosis evasion, dedifferentiation, metastasis, and therapy resistance. For instance, loss of tumor suppressor TFs such as TP53, in combination with KRAS mutations or MYC amplification, amplifies oncogenic signaling and enables immune evasion.

A subset of TFs, termed tumor-driving master TFs, are aberrantly expressed across multiple cancer types. Normally essential for embryonic development and lineage specification, these factors are co-opted by cancer cells to sustain malignant transcriptional networks. For instance, in leukemia, dysregulation of TFs such as RUNX1, PML, BCL6, MYC, E2A, and NOTCH1 is frequently observed. Similarly, TFs linked to neural development including MYCN, PHOX2B, SOX10, and GATA3, drive high-risk neuroblastoma (NB). The tumor suppressor p53, crucial for maintaining genomic integrity during development, is often mutated in osteosarcoma and rhabdomyosarcoma (RMS), two common pediatric solid tumors. In pediatric cancers, where developmental programs remain intrinsically active, TF dysregulation exerts profound oncogenic effects.

Aberrant TF activity can block differentiation, maintain stem-like states, and accelerate proliferation, thereby contributing to the aggressive nature of many childhood malignancies, including leukemia, lymphoma, and central nervous system (CNS) tumors. Given the rapid growth kinetics of developing tissues, disruption of developmental TF networks in children may have amplified oncogenic consequences relative to adult cancers. Mechanistically, TF dysregulation in pediatric tumors arises through chromosomal rearrangements (e.g., *ETV6::RUNX1* in acute lymphoblastic leukemia (ALL), *EWS::FLI1* in Ewing sarcoma), overexpression (*MYCN* in NB, *GLI1* in Sonic Hedgehog (SHH) medulloblastoma (MB)), or mutations (*PHOX2B*, *TP53*, *MYOD1* in spindle cell RMS). Aberrant activation of signaling cascades such as IGF-1R or PI3K/AKT/mTOR further amplifies TF-driven oncogenic transcriptional programs.

Over the past decade, significant progress has been made in targeting these TFs or their upstream regulators that drive their activation. Small-molecule inhibitors have been developed to directly target TFs, including PROTACs. Other strategies aim to disrupt their DNA binding, interactions with partner proteins, transcriptional regulators, or epigenetic modulators such as histone deacetylases (HDACs). Additionally, blocking the nuclear export of TFs has emerged as a viable therapeutic approach. Pediatric tumors driven by fusion TFs, such as *EWS::FLI1*, *PAX3::FOXO1*, and *RUNX1* fusions, are key targets for these therapies. Furthermore, combination strategies integrating TF inhibitors with epigenetic modulators are gaining traction as a promising treatment approach. Despite these advancements, clinical trials specifically targeting TFs in pediatric cancers remain fewer compared to those focused on adult cancers. This disparity is likely due to the lower incidence of pediatric tumors, combined with the greater emphasis on developing therapies for adult malignancies.

However, cancer is now the leading cause of death from disease in children, highlighting the need to focus on improving pediatric cancer treatments. As a result, there is a pressing need for more targeted research in this area to ensure that pediatric cancer patients benefit from these promising therapeutic strategies. Therefore, this review will highlight the critical role of TFs in pediatric tumor development and examine the emerging approaches being explored to develop small-molecule modulators that can target and regulate these essential proteins.

## 2. Chromosomal Rearrangements in TFs

A hallmark of pediatric malignancies: Chromosomal rearrangements involving TFs, such as translocations, inversions, deletions, and amplifications, are common in pediatric cancers. These alterations often result in constitutively active genes that drive uncontrolled cell growth and tumor development.

### 2.1. Chromosomal Translocations

Translocations that generate fusion oncogenic TFs, resulting from the joining of DNA from different chromosomes, are common in leukemias and sarcomas. In ALL, which accounts for 80% of pediatric leukemia cases, various translocations involving fused TFs contribute to transforming normal blood cells into leukemic cells. The most common is *ETV6::RUNX1* (t(12;21)(p13;q22)) translocation, where the transcriptional repressor *ETV6* fuses with the TF *RUNX1*, disrupting its normal role in hematopoietic differentiation and promoting leukemogenesis. Although *ETV6::RUNX1* ALL was once seen as a uniform, favorable-risk subtype, recent transcriptomic analysis of 194 pediatric cases has identified two distinct subtypes, C1 and C2.

C1 is more common in older patients, exhibiting a stem-like gene expression profile and increased chemotherapy resistance. C2, found mainly in younger children, is more sensitive to drugs and shows better early treatment responses. Genetically, C2 is strongly linked to *PAX5* deletions and *RAS* mutations, while C1 has fewer of these alterations. Functional studies showed that restoring *PAX5* expression in C2 lowers CDK6 levels, slows the cell cycle, and reduces sensitivity to chemotherapy. These findings highlight that *ETV6::RUNX1* ALL consists of biologically distinct subgroups with different therapeutic vulnerabilities [4].

Expanding upon this molecular framework, a recent multiomic study further revealed that therapy response in *ETV6::RUNX1* ALL depends on specific genomic features present at diagnosis [5]. Fast responders tend to carry *APOBEC* mutational signatures and high expression of cell cycle-related genes, while slow responders display productive IGK rearrangements, indicating a more mature pre-B cell origin. Mutations in *INTS1*, *NF1*, and *TP53* are more common in slower responders. Importantly, copy number alterations on chromosome 12p, specifically deletions affecting *ETV6*, *KRAS*, and *FKBP4*, were more common in fast responders and linked to increased drug sensitivity, while gains in this region involving the derivative chromosome der(21)t(12;21) were associated with drug resistance. These findings suggest that *ETV6::RUNX1* ALL is a molecularly heterogeneous disease with subtypes and genomic factors that influence therapy response and prognosis [5].

Similarly, translocations involving the histone methyltransferase *KMT2A* (also known as MLL or Mixed-lineage leukemia), located at 11q23, lead to fusion proteins with various partners that disrupt normal blood cell development. These KMT2A fusion proteins are found in about 75% of infant ALL, 5% of childhood ALL, and 18% of pediatric ALL, and are often linked to poor outcomes. The most common KMT2A fusion partners include *AF4*, *AF9*, *ELL*, and *ENL (MLLT1)*, located on chromosomes 4q21, 9p22, 19p13.1, and 19p13.3, respectively. These result in the translocations t(4;11)(q21;q23), t(9;11)(p22;q23), t(11;19)(q23;p13.1), and t(11;19)(q23;p13.3). Although these fusion partners are not TFs, they are key components of the transcriptional machinery. Specifically, they are part of the Super Elongation Complex (SEC), which plays a crucial role in regulating the release of paused RNA polymerase II into productive transcription elongation [6]. The SEC functions at the transcriptional elongation checkpoint control (TECC) stage, ensuring proper transcriptional progression and preventing premature termination of gene expression [6,7]. *KMT2A* fusions are accompanied by additional TF mutations, which further contribute to relapse and poor prognosis. For instance, whole-exome sequencing of a patient who relapsed six months after remission from *KMT2A::MLLT3* rearranged acute monocytic leukemia (later progressing to ALL) revealed two somatic mutations in the TF *PAX5* [8].

*PAX5*, often referred to as the “guardian of B-cell identity”, is a lineage-specific TF essential for B-cell development and differentiation. PAX5 belongs to the Paired box (PAX) gene family, which is a group of TFs essential for development, cell lineage specification, and organogenesis [9]. Various chromosomal translocations involving *PAX5* result in fusion genes that disrupt its regulatory function, contributing to leukemogenesis. Notable *PAX5* fusion partners identified in pediatric B-cell precursor ALL (BCP-ALL) include: *PAX5::PML* (t(9;15)(p13;q24)), *PAX5::KIAA1549L* (t(9;11)(p13.2;p12)), *PAX5::ELN* (t(7;9)(q11;p13)), *PAX5::AUTS2* (t(7;9)(q11.2;p13.2)), *PAX5::FOXP1* (t(3;9)(p13;p13)), and *PAX5::ETV6* (t(9;12)(p13;p13)) rearrangements. These fusions create dominant negative *PAX5*, which impairs PAX5 ability to regulate gene expression, disrupting normal B-cell differentiation and contributing to leukemic transformation.

Other members of the *PAX* family, including *PAX3* (on chromosome 2) and *PAX7* (on chromosome 1), are involved in translocations with the growth-promoting TF *FOXO1* (on chromosome 13q14) in alveolar RMS, a highly aggressive pediatric soft tissue sarcoma. These translocations result in t(2;13)(q35;q14) and t(1;13)(p36;q14), generating *PAX3::FOXO1* and *PAX7::FOXO1* fusion genes, respectively [10,11]. While reciprocal fusions (e.g., *FOXO1::PAX3* or *FOXO1::PAX7*) may also occur, they are not typically expressed and do not contribute to tumorigenesis. These translocations represent early, defining events in alveolar RMS tumorigenesis, occurring more frequently than any other genetic lesion in the disease. *PAX3::FOXO1* is detected in ~61% of cases, while *PAX7::FOXO1* is found in ~15% [12]. Compared to *PAX7::FOXO1*, *PAX3::FOXO1* is associated with higher rates of metastasis (54.4% vs. 23.5%), more aggressive biology, and worse clinical outcomes due to its stronger transcriptional activity and oncogenic potential [12,13].

Similarly, pediatric sarcomas are often driven by translocations that generate chimeric TFs with gain-of-function properties. A classic example is the *EWSR1::FLI1* fusion in Ewing sarcoma, which is caused by the t(11;22)(q24;q12) translocation. This translocation fuses the N-terminal domain of the *EWSR1* transcriptional co-activator (located on chromosome 22) with the *FLI1* TF, resulting in a potent oncogenic TF that abnormally activates FLI1 target genes. FLI1, a member of the ETS family of TFs, regulates cell growth, differentiation, and development. Other members of the ETS family are also involved in pediatric sarcomas through similar rearrangements. For example, the *ERG* gene is rearranged in 5–10% of cases via the t(21;22)(q22;q12) translocation, leading to an *EWSR1::ERG* fusion. In rarer instances, *EWSR1* fuses with other ETS TFs, including *ETV1*, *E1AF*, *and FEV*, through the t(7;22)(p22;q12), t(17;22)(q12;q12), and t(2;22)(q33;q12) translocations, respectively. In all these cases, the *EWSR1* gene contributes its strong N-terminal transcriptional activation domain, acting as a booster that makes the fusion protein hyperactive in driving gene expression.

Fused, hyperactive ETS family TFs are key genomic hallmarks in certain pediatric brain tumors, where they aberrantly activate oncogenic processes. A prominent example is the *ETV6::NTRK3* translocation in infantile glioblastoma and other high-grade gliomas. It creates the *ETV6::NTRK3* oncogenic fusion gene due to the t(12;15)(p13;q25) translocation. In this fusion, *ETV6*, a transcriptional repressor located on chromosome 12 and a member of the ETS family, fuses with *NTRK3*, which encodes TrkC, a receptor tyrosine kinase on chromosome 15. The resulting *ETV6::NTRK3* fusion protein combines the self-dimerization domain of *ETV6* with the tyrosine kinase domain of *NTRK3*, leading to constitutive activation of the TrkC kinase. This activation drives oncogenic signaling through the RAS/MAPK, PI3K/AKT, and STAT pathways, contributing to tumorigenesis. Several novel NTRK fusions are also observed in rare tumor types, including *KCTD16::NTRK1* in ganglioglioma and *IRF2BP2::NTRK3* in papillary thyroid carcinomas (Table 1).

### 2.2. Chromosomal Inversions

Chromosomal inversions occur when a segment of DNA breaks and reinserts in reverse orientation within the same chromosome. This can generate fusion genes or reposition regulatory elements, such as enhancers or silencers, near unrelated genes. These alterations may lead to the production of oncogenic fusion proteins or the inactivation of tumor suppressors, promoting abnormal cell growth. Though less common, TFs involving inversions are present in certain cases of pediatric AML, ALL, Ewing sarcoma, and infantile fibrosarcoma. In the AML M4 subtype with eosinophilia (AML M4Eo), which occurs in 5–9% of AML cases, a pericentric inversion inv(16)(p13q22) fuses *CBFB* (core-binding factor, beta subunit) at 16q22 with *MYH11* at 16p13, forming the oncogenic *CBFB::MYH11* fusion gene. CBFB is the β subunit of the core-binding TF complex in the PEBP2/CBF family.

Although CBFB does not bind DNA directly, it enhances the DNA-binding ability of the α subunit, RUNX1. Together, this CBF complex binds core sites within enhancers and promoters of genes essential for hematopoiesis. The *CBFB::MYH11* fusion protein disrupts normal hematopoietic differentiation through multiple mechanisms: it interferes with TF complex assembly, sequesters RUNX1 in the cytoplasm, and recruits HDACs to repress gene expression. It also promotes the expression of genes involved in hematopoietic stem cell self-renewal, such as *ID1*, *LMO1*, and *JAG1* [67]. As a result, RUNX1 is unable to bind DNA and activate transcription of target genes required for the maturation of monocytic and eosinophilic lineages, leading to a differentiation block and the accumulation of immature myeloid precursors (myeloblasts).

Other less common but high-risk inversions occur in 2–3% of pediatric AML cases. These include the *GATA2::MECOM* (EVI1) fusion resulting from inv(3)(q21q26.2) or t(3;3)(q21;q26.2), and the *CBFA2T3::GLIS2* fusion from inv(16)(p13.3q24.3). The *GATA2::MECOM* fusion relocates a distal enhancer of the *GATA2* TF near *MECOM*, leading to MECOM overexpression and suppression of GATA2. This dysregulation blocks myeloid differentiation and promotes the proliferation of HSCs and myeloid leukemia cells. The *CBFA2T3::GLIS2* fusion creates a chimeric TF that impairs normal hematopoietic differentiation, contributing to leukemogenesis.

In ALL, rare inversions are also associated with poor prognosis and therapy resistance. A notable example is inv(14)(q11q32), which leads to aberrant overexpression of *TCL1*, a gene frequently upregulated in T- and B-cell lymphomas and various solid tumors [68,69]. While inv(14)(q11q32) is more commonly seen in T-cell chronic lymphoproliferative disorders, it is rarely observed in T-ALL but remains significant due to its association with high chemoresistance [68,69]. Inversions also act as contributing factors to various outcomes in Ewing sarcoma tumors. Although the most common genetic alterations in Ewing sarcoma involve translocations in *EWS* gene and *ETS* family of TFs, inversions play a significant role in prognosis and contribute to the overall complexity of the karyotype. One such example is a reciprocal insertion/inversion between chromosomes 21 and 22, involving the excision and exchange of 3′ *ERG* and 3′ *EWS* segments, leading to the formation of the oncogenic *EWS::ERG* fusion gene [70] (Table 1).

### 2.3. Chromosomal Deletions

Chromosomal deletions involve the loss of a chromosome segment, which can disrupt essential genes, including tumor suppressors. This disruption can impair gene function, promoting cancer initiation and progression. In pediatric solid tumors, such deletions significantly affect disease development and prognosis. Recurrent losses in regions like 1p, 11q, 11p and 17p are associated with higher risk and worse outcomes in NB, Wilms tumor, and MB. For instance, deletions of 11q and 1p36 (up to 65% of high-risk cases) are common in NB and are linked to impaired differentiation and aggressive tumor behavior. A key TF near the 11q deletion breakpoint is *PHOX2A*, which plays a critical role in maintaining noradrenergic neuronal differentiation during embryonic development. *PHOX2A* expression is reduced in unfavorable NB, suggesting that its loss due to 11q deletion may contribute to aggressive disease.

However, some studies report that *PHOX2A* is not mutated but overexpressed in several NB tumors and cell lines, indicating a complex role in tumor biology [71]. The 11q deletion can also reposition proximal genes near active TFs, leading to constitutively active fusion transcripts. In NB, genes located at the proximal side of the 11q deletion, such as *MLL* and *PAFAH1B2*, have been found fused to the TF *FOXR1*, forming *MLL::FOXR1* and *PAFAH1B2::FOXR1* fusions [72]. *FOXR1* is recurrently activated in NB, and its fusion with these genes may drive tumor proliferation through aberrant activation of oncogenic pathways [72].

Several tumor suppressor genes located on chromosome 11q, such as *DLG2* and *ATM*, are frequently lost or repressed in NB, contributing to tumor progression. Notably, these genes can also be transcriptionally silenced by oncogenic TFs, even in the absence of genetic deletions, thereby maintaining an undifferentiated, proliferative state in tumor cells. For example, ALK signaling has been shown to repress *DLG2* expression through the TF SP1, thereby sustaining the undifferentiated identity of neural crest-derived progenitor cells [73]. Similarly, MYCN, a TF often amplified in NB, along with E2F1, contributes to the transcriptional repression of *ATM*. MYCN promotes the expression of miR-421, a microRNA that targets the 3′ untranslated region (UTR) of *ATM* mRNA, leading to decreased ATM protein levels and enhancing tumor cell survival and genomic instability [74]. This mechanism of transcriptional repression by oncogenic TFs such as MYCN, SP1, and E2F1 mimics the functional consequences of chromosomal loss, effectively inactivating tumor suppressor pathways. The convergence of chromosomal deletions and transcriptional silencing compounds the oncogenic potential of NB and reflects the multilayered complexity of its genetic and epigenetic regulation.

Moreover, *MYCN* amplification in NB is frequently associated with 1p36 deletions, observed in approximately 70% of cases. These deletions result in the loss of tumor suppressor TFs such as *ARID1A*, a chromatin remodeler known to constrain *MYCN*-driven transcriptional programs. Loss of *ARID1A* facilitates the oncogenic activity of *MYCN*, further amplifying proliferative and survival signals in NB cells [75]. These findings emphasize that NB pathogenesis is not solely the result of discrete genetic events but arises from an intricate interplay between structural chromosomal alterations and TF-mediated gene regulation. This layered mechanism of gene inactivation illustrates the complex regulatory architecture of high-risk NB. It underscores the need for integrative therapeutic strategies that target both genetic lesions and the epigenetic or transcriptional machinery driving tumor progression.

Another important tumor suppressor TF affected by deletions at chromosome 11p13 is Wilms tumor 1 (*WT1*), which is essential for normal kidney development. *WT1* mutations, including deletions and point mutations, are observed in approximately 10–20% of Wilms tumors, implicating their loss in a subset of cases [76,77]. *WT1* encodes a DNA and RNA-binding protein that regulates nephron progenitor differentiation and lineage progression. It supports the survival of multipotent metanephric mesenchyme (MM) cells by inducing FGF signaling and repressing bone morphogenetic protein (BMP)/pSMAD signaling. WT1 directly activates Fgf16 and Fgf20 expression, which drive MM cell proliferation and suppress apoptosis by antagonizing pSMAD activity. In the absence of WT1, reduced FGF signaling and heightened BMP/pSMAD activity lead to apoptosis in the developing metanephric blastema, impairing nephrogenesis [78].

In addition, WT1 regulates structural genes crucial for podocyte development, including nephrin and podocalyxin, which are expressed in podocytes derived from MM during kidney morphogenesis [79,80]. WT1 promotes nephrin transcription by binding to a specific element in its promoter, while its regulatory interaction with podocalyxin remains less defined. Mouse models lacking *WT1* exhibit reduced expression of both genes, resulting in defective podocyte differentiation and nephron formation [79,80,81]. Beyond kidney development, WT1 also plays a role in pediatric and adult T-ALL. WT1 is crucial for the early stages of blood cell formation, and its expression is largely restricted to primitive hematopoietic progenitors, particularly those with the CD34+ and CD38− phenotype [82].

*WT1* mutations or deletions are found in approximately 10% of pediatric and adult T-ALL cases. It consists of heterozygous frameshift insertions or deletions that introduce premature stop codons, resulting in truncated proteins lacking the DNA-binding domain or triggering nonsense-mediated decay [83,84]. *WT1* loss in T-ALL impairs transcriptional activation of *TP53* and elevates pro-survival genes *BIRC5* (survivin), *XIAP*, and *HMOX2* (HO-2) following DNA damage, thereby promoting leukemic cell survival and treatment resistance [85]. Furthermore, deletions involving the short arm of chromosome 1 (1p) are frequently observed structural abnormalities in pediatric brain tumors, including astrocytomas and primitive neuroectodermal tumors (PNETs)/MBs. Breakpoints can range from 1p1 to 1p3. Such deletions are less common in adult brain tumors, highlighting a pediatric-specific pattern (Table 1).

### 2.4. Chromosomal Amplifications

Chromosomal amplification refers to the abnormal duplication of a chromosome segment, resulting in multiple copies of the same DNA sequence and leading to gene overexpression. The most well-known example is *MYCN* amplification in NB, which occurs in about 22–25% of cases [86]. This amplification is strongly linked to advanced-stage disease, rapid tumor growth, and poor clinical outcomes. *MYCN* amplification is a key oncogenic driver and an important factor in risk stratification for NB patients.

Amplified *MYCN* accelerates cell cycle progression through several mechanisms. It upregulates CDK4, which competes with CDK2 for binding to the tumor suppressor p21, reducing p21 availability and allowing for unchecked CDK activity [87]. MYCN also represses anti-proliferative genes such as *DKK1* (Dickkopf-1), which inhibits WNT/β-catenin signaling, and *CDKL5*, which halts cells at the G0/G1 phase [88,89]. Additionally, MYCN directly activates *ID2*, a helix-loop-helix transcription factor that inactivates the RB protein, further promoting cell cycle progression [90]. MYCN binds to the E-box motif in the promoter of *E2F5* TF, enhancing its transcription, and also upregulates TFAP4, which regulates genes like *PRPS2* and *SDC1* involved in proliferation and metastasis [91,92].

At the epigenetic level, MYCN interacts with DNA methyltransferase DNMT3A to silence transcription of p21Cip1, and upregulates histone deacetylases HDAC2 and SIRT1 [93]. HDAC2 suppresses cyclin G2, a negative regulator of the cell cycle, while SIRT1 represses MAPK phosphatase 3, sustaining ERK activation [94,95]. MYCN also recruits HDAC2 to the miR-183 promoter, repressing its expression. HDAC2 inhibition restores H4 acetylation, activates miR-183, induces apoptosis, and reduces anchorage-independent growth [96]. MYCN-driven transcription further relies on the KAT module of the SAGA complex, which includes the acetyltransferase GCN5. This module regulates histone acetylation to sustain MYCN oncogenic gene expression. Disrupting the KAT module impairs MYCN activity and tumor growth, making it a promising therapeutic target in MYCN-amplified NB [97].

Beyond NB, *MYCN* amplification is also observed in pediatric brain tumors, including pediatric high-grade gliomas (pHGGs) and MBs with SHH, Group 3, and Group 4 subtypes [98,99]. pHGGs are a diverse group classified into four molecular types: H3.3-mutant, IDH-mutant, H3.3 and IDH wild-type, and infant-type hemispheric glioma [100]. *MYCN* amplification is most commonly associated with the H3.3 and IDH wild-type subgroup, which includes a distinct molecular subtype known as pHGG-MYCN. This subgroup is defined by high-level *MYCN* amplification and a unique DNA methylation signature [101]. It is part of a broader molecular classification that also includes pHGG-RTK1 and pHGG-RTK2, characterized by receptor tyrosine kinase gene alterations. HGG-MYCN tumors are considered a distinct and aggressive entity, often harboring co-alterations in *TP53* and *MYCN*, and are associated with particularly poor clinical outcomes [101]. Although the global incidence of pHGGs is relatively low (1.1–1.78 cases per 100,000 children), they account for over 40% of pediatric brain tumor deaths, with HGG-MYCN representing the subgroup with the worst survival [102,103].

In MB, about 7% of SHH subgroup patients show a gain of chromosome 2, which contains the *MYCN* gene, resulting in its amplification [104,105]. In Group 3, *MYCN* amplification occurs in 2–4% of cases, while in Group 4, it is found in about 6% [106,107]. Group 4 tumors are often characterized by *MYCN* amplification and isochromosome 17q [108,109]. In contrast, *MYC* (amplification is more common in Group 3, occurring in 10–17% of patients [110]. *MYC* amplification is a strong indicator of poor prognosis and worse outcomes in both groups. The presence of *MYC* or *MYCN* amplification, along with alterations such as *TP53* mutations, *MYCL1* amplification (especially in SHH), and mutations in histone methyltransferases like *MLL2* and *MLL3*, is linked to inferior survival, aggressive tumor behavior, and resistance to therapy [111,112,113].

Furthermore, *MYCN* is also amplified in alveolar RMS (25–43% of cases), retinoblastoma (2%), and Wilms tumors (12.7%) and is strongly associated with poor outcomes [114,115,116,117]. In fusion-positive alveolar RMS, approximately 20% of cases show *MYCN* amplification due to genomic gain at chromosome 2p24 [118]. Amplified *MYCN* cooperates with *PAX3::FOXO1* and *PAX7::FOXO1* fusions to promote cell proliferation, block differentiation, and drive malignant progression [119]. Other amplified TFs in pediatric tumors include *OTX2*, commonly amplified via duplication of 14q22.3 in MB; *Twist2*, amplified in RMS; and *PLAGL1/PLAGL2*, amplified in CNS embryonal tumors, all of which contribute to tumor development and progression [120,121,122] (Table 1). Figure 1 summarizes common chromosomal rearrangements observed in various pediatric cancers.

### 2.5. Complex Structural Variations (SVs)

Complex SVs are large, intricate changes in DNA that involve multiple pieces being rearranged, deleted, duplicated, or inserted. Unlike simple alterations such as a single translocation or deletion, complex SVs feature multiple breakpoints occurring simultaneously, often in one event, leading to a tangled pattern of DNA modifications. These changes can impact several genes or chromosome regions at once and are frequently observed in cancer genomes, where they can promote disease progression. Due to their complexity and size, they are more difficult to detect and interpret than simple rearrangements [123].

Recent genomic research has revealed that complex SVs, including chromothripsis (shattering of one or more chromosomes followed by random reassembly), chromoplexy (a chain-like series of translocations and deletions linking multiple chromosomes), and extrachromosomal DNA formation (circular DNA fragments carrying oncogenes that replicate independently of chromosomes) are common in pediatric solid tumors and significantly influence tumor evolution and progression. In a study of 120 primary tumors, including NB, Ewing sarcoma, Wilms tumor, hepatoblastoma, and RMS, approximately 47% exhibited chromosomal rearrangements. These rearrangements often target key driver genes or lead to unfavorable chromosomal alterations, contributing to tumor development and advancement.

Examples include *MYCN* amplifications driven by extrachromosomal DNA in NB and *EWSR1::FLI1* fusions caused by chromoplexy in Ewing sarcoma. The presence of these complex genomic rearrangements (CGRs) correlates with poorer clinical outcomes, suggesting they could be useful for risk assessment or targeted treatment strategies. The data further indicate that CGRs likely originate early in tumor evolution as single catastrophic events and are shaped by positive and negative selection within the tumor genome [124].

## 3. Point Mutations

Unlike large structural changes during chromosomal rearrangements, point mutations are small-scale alterations at the nucleotide level, involving substitution, insertion, or deletion of one or a few bases. While chromosomal rearrangements affect gene structure and genomic organization, point mutations alter the DNA sequence within a gene. Pediatric cancers exhibit a spectrum of point mutations primarily in developmental and epigenetic regulators, signaling pathway genes, and fusion-associated partners that collaboratively drive tumorigenesis.

One example is *CTNNB1* mutations, which encode the β-catenin protein in MB. These mutations frequently occur in exon 3, targeting phosphorylation sites such as Ser33, Ser37, Thr41, and Ser45 that regulate β-catenin stability. Loss of these phosphorylation sites prevents β-catenin degradation, resulting in its nuclear accumulation and constitutive activation of the WNT/β-catenin signaling pathway. This leads to aberrant transcription of target genes that promote tumorigenesis in WNT-activated MB, which represents approximately 10% of MB cases [125]. Specific mutations, such as p.Ser37del (ΔS37) in the *CTNNB1* gene, confer gain-of-function properties by enhancing TCF/LEF-driven transcription, contributing to early tumor initiation [126].

Another example involves *DDX3X*, an RNA helicase gene frequently mutated in MB. Most alterations in the gene are single-point amino acid substitutions that yield catalytically impaired DDX3X proteins. These mutant proteins disrupt translation initiation and globally reduce protein synthesis [127]. *DDX3X* mutations often co-occur with *CTNNB1* mutations, suggesting cooperative roles in WNT-driven MB pathogenesis [128]. Moreover, the *BRAF V600E* mutation, a single nucleotide change (c.1799T>A) that substitutes valine with glutamate at position 600, is found in various pediatric CNS gliomas. This mutation leads to constitutive activation of BRAF kinase that drives the downstream MAPK signaling pathway independently of RAS regulation [129].

Mutations in *KRAS*, *PIK3CA*, and *TP53* also occur across pediatric cancers, although they may arise later or exist as subclonal events. Activating missense mutations in *PIK3CA*, such as E545K and H1047R, enhance PI3K pathway signaling and are detected in several pediatric solid tumors [130]. *TP53* missense mutations, which typically affect the DNA-binding domain, are recurrent in high-grade gliomas, MB, pediatric adrenocortical carcinoma, and leukemias, leading to loss of tumor suppressor function and impaired genomic stability [131,132,133,134].

In pediatric ALL, both missense and truncating mutations in *CREBBP* (and, less frequently, *EP300*) target the histone acetyltransferase (HAT) domain, impairing acetyltransferase activity and thereby disrupting chromatin acetylation and transcriptional regulation [135,136]. Similarly, *ETV6* is frequently altered by point mutations (missense or frameshift) that compromise its transcriptional repressor function, contributing to leukemogenesis [137]. In T-ALL, aberrant *TAL1* expression is often driven by somatic mutations in noncoding regulatory regions, such as microinsertions or enhancer-activating mutations near the *TAL1* locus. These noncoding alterations create de novo super-enhancers that sustain ectopic TAL1 expression, which is normally silenced during T-cell maturation. TAL1 interacts with cofactors including LMO1/2, GATA3, and RUNX1, forming an autoregulatory transcriptional loop that maintains its own expression and disrupts normal T-cell differentiation, thereby promoting malignant transformation [138,139].

In summary, point mutations in pediatric cancers affect crucial genes involved in developmental processes, epigenetic regulation, and signaling pathways, leading to dysregulated cellular growth and tumor development.

## 4. TFs as Therapeutic Targets in Pediatric Cancers

TFs have historically been considered difficult to target therapeutically due to their lack of enzymatic activity and structurally disordered regions, which complicate traditional drug design [140]. However, recent technological and pharmacological advances, such as small-molecule inhibitors, PROTACs, and structure-based drug design, have significantly improved the feasibility of targeting TFs. These innovative strategies are showing encouraging results in preclinical studies and early-phase clinical trials. This underscores their potential as effective therapeutic options in both adult and pediatric cancers.

Despite these advances, the translation of novel therapies to pediatric oncology remains slow. A substantial gap exists between drug development for adults and children [141]. The median time from a first-in-human trial to the corresponding first-in-child trial is 6.5 years, with a range extending up to 27.7 years. Moreover, between 1997 and 2017, only 5.1% of oncology drugs initially approved by the U.S. Food and Drug Administration (FDA) included pediatric patients in the approval process [141]. These statistics underscore the urgent need for dedicated pediatric drug development strategies to accelerate access to emerging therapies, including those targeting TFs. To date, the most clinically validated and widely used TF-targeting agents in pediatric oncology are glucocorticoid receptor (GR) agonists [142]. These agents have been highly effective in treating pediatric ALL and are associated with improved survival outcomes [142].

GR-targeting drugs constitute 29% of all FDA-approved drugs that target nuclear TFs [143]. These agonists bind to the glucocorticoid receptor (GR) in the cytoplasm, inducing conformational changes that facilitate GR translocation into the nucleus. In the nucleus, the GR complex binds to glucocorticoid response elements (GREs) in the DNA, triggering the expression of pro-apoptotic genes (Bcl-2 family members) and repressing genes involved in cell survival and proliferation [144,145].

In addition to directly regulating gene expression through DNA binding, GR also modulates other signaling pathways via protein–protein interactions with specific TFs, mediated by its single receptor protein module, GRα [146]. These interactions influence the transcriptional activity of various target genes, either enhancing or suppressing their expression depending on the cellular context. GR agonists such as prednisone, prednisolone, and dexamethasone are commonly used in combination with other chemotherapeutic agents to eliminate leukemia cells and manage inflammation and allergic reactions during treatment. Their success illustrates both the feasibility and therapeutic impact of TF-targeted drugs in pediatric cancers. This highlights the promise of expanding such strategies to other transcriptional regulators.

However, over time, leukemia can develop resistance to GR agonists, which presents a major barrier to effective treatment and is associated with significantly poorer clinical outcomes [147]. Several mechanisms contribute to this resistance. These include reduced GR expression due to deletions or mutations in the *NR3C1* gene (located at 5q31.3), and loss or mutation of *IKZF1* (encoding the Ikaros TF), which disrupts glucocorticoid-induced gene expression [148,149,150,151]. Additionally, downregulation of GR downstream pro-apoptotic targets, such as Bax, or upregulation of anti-apoptotic proteins like BCL-2 and MCL1, can impair the apoptotic response [152,153,154,155]. Other mechanisms include FBXW7-mediated ubiquitylation and proteasomal degradation of GR, reducing its stability and nuclear activity [156]. Furthermore, the activation of signaling pathways such as IL-7R, PI3K/AKT, and mTOR has been strongly linked to glucocorticoid resistance, as these pathways promote cell survival and proliferation, counteracting the apoptotic effects of GR activation [157,158,159,160].

One strategy to overcome resistance to GC agonists involves the use of proteasome inhibitors. Agents such as bortezomib, a reversible inhibitor of the β5 chymotrypsin-like subunit of the 20S proteolytic core of the 26S proteasome, and carfilzomib, an irreversible inhibitor of the β5/β5i subunits and a second-generation proteasome inhibitor, have shown potential in this context. By inhibiting proteasomal degradation, these agents increase the accumulation of functional GR within cells, thereby enhancing GR-mediated transcriptional activity and potentially restoring glucocorticoid sensitivity [150,161,162,163]. Bortezomib, in combination with prednisone, has been evaluated in Phase I, II, and III clinical trials in pediatric patients with relapsed ALL, T-lymphoblastic lymphoma (T-LL), T-ALL, pre-B ALL, and B-lymphoblastic lymphoma (B-LL) [164,165,166]. Similarly, carfilzomib combined with dexamethasone has been tested in Phase I/II studies for children with relapsed or refractory ALL. This combination offers a promising avenue to overcome resistance and improve therapeutic outcomes (ClinicalTrials.gov ID: NCT02303821).

Another innovative approach used by St. Jude Children’s Research Hospital involves molecular glue degraders. These are small molecules that bring a target protein into proximity with an E3 ubiquitin ligase, leading to its ubiquitination and subsequent proteasomal degradation [167] (Figure 2). St. Jude has built a large library of rationally designed molecular glues that specifically recruit the E3 ligase substrate receptor CRBN (cereblon). CRBN serves as a substrate receptor within the CRL4^CRBN^E3 ligase complex that marks target proteins for proteasomal degradation [168]. It is also the molecular target of immunomodulatory drugs (IMiDs) such as thalidomide, lenalidomide, and pomalidomide. These drugs alter CRBN’s substrate specificity, enabling it to recruit and degrade new targets. Two key lymphoid TFs, IKZF1 and IKZF3, were among the first neosubstrates identified for IMiDs, providing a mechanistic explanation for the clinical efficacy of these agents in treating multiple myeloma [169,170].

Building on this platform, SJ7095 was identified by the Zoran Rankovic group at St. Jude through high-throughput screening of a CRBN ligand library across a panel of patient-derived pediatric and adult cancer cell lines [167]. SJ7095 functions as a potent molecular glue degrader that promotes CRBN-dependent degradation of casein kinase 1α (CK1α), IKZF1, and IKZF3. This results in strong antiproliferative effects in cancer cells [167]. Furthermore, DEG-35 and DEG-77 are cereblon-dependent degraders targeting IKZF2 and CK1α that have been shown to delay leukemia progression in both murine and human AML models [171]. Another notable example is CC-90009, which selectively degrades GSPT1, a translation termination factor that facilitates the release of the completed polypeptide from the ribosome. CC-90009 has shown promising activity in vitro and in vivo by triggering apoptosis in AML cells and eliminating leukemic stem cells [172]. Although it was evaluated in a phase I trial for relapsed or refractory AML and high-risk MDS in adults, the study was terminated early due to limited clinical efficacy (ClinicalTrials.gov ID: NCT02848001). The molecular glue degrader approach is rapidly growing as an effective strategy to target traditionally undruggable TFs across various cancer types. Emerging targets include oncogenic regulators such as BRD4, CTNNB1 (β-catenin), and NFKB1 (NF-κB). Ongoing preclinical and clinical efforts continue to expand the therapeutic potential of targeted protein degradation [173,174,175] (Table 2).

Another related approach involves PROTACs, which also utilize the cell’s ubiquitin-proteasome system (UPS) to selectively degrade proteins (Figure 2). Unlike molecular glues that reprogram existing interactions, PROTACs are bifunctional molecules made of two ligands linked together. One ligand binds the target protein, while the other attaches to an E3 ubiquitin ligase, bringing the two close together. This promotes the ubiquitination and subsequent degradation of the target protein by the proteasome. Ongoing clinical and preclinical studies are exploring the potential of PROTACs to selectively eliminate oncogenic TFs in childhood cancers. For example, a synthetic DNA oligonucleotide-based PROTAC, d(GGAA)_3_s, has been developed to selectively degrade ETV6 in Ewing sarcoma cells [196]. This PROTAC contains three consecutive GGAA repeat motifs, which specifically bind to the ETS DNA-binding domain of ETV6. ETV6 competes with the oncogenic fusion protein EWS/FLI-1 for binding to short GGAA repeat sequences. By binding to ETV6, d(GGAA)_3_s facilitates its ubiquitination and proteasomal degradation, leading to suppressed cell proliferation and increased sensitivity to chemotherapy in Ewing sarcoma models [196]. Since ETV6 is also implicated in leukemias and lymphomas, this strategy may have broader applications in pediatric cancers.

The PROTAC degraders are also being developed for other pediatric cancers that involve specific pathogenic proteins or fusion drivers. Examples include PROTACs such as GU3341 that target CDK6 and GSPT1. Downregulation of GSPT1 suppresses tumor growth, causes cell cycle arrest, and triggers apoptosis in pediatric AML cells with *RUNX1::RUNX1T1* or *FUS::ERG fusions* [197]. The preclinical success of GSPT1-targeting PROTACs continues to generate interest in their potential applications, especially in pediatric AML and other cancers [198,199]. Another example is SJ11646, a highly effective PROTAC designed to degrade LCK, a crucial kinase involved in T-ALL [200]. This PROTAC was created by linking the FDA-approved kinase inhibitor dasatinib to a phenyl-glutarimide cereblon ligand, enabling selective degradation of LCK through the ubiquitin-proteasome system. SJ11646 is over 1000 times more cytotoxic and has more durable effects than dasatinib in LCK-dependent T-ALL cell lines, patient-derived samples, and mouse xenograft models. It also maintains strong binding affinity to other oncogenic kinases, such as ABL1, KIT, and DDR1, indicating potential usefulness beyond T-ALL [200]. Additional examples are summarized in Table 2.

Most PROTAC clinical trials so far focus on adult cancers, including prostate, breast, lung, and hematologic malignancies, with several PROTACs targeting proteins like the androgen receptor, BTK, STAT3, and Bcl-xL progressing to phase I/II studies. In contrast, pediatric studies remain mostly at the preclinical and translational research stage. While no large-scale pediatric cancer clinical trials of PROTACs have yet been reported, early clinical successes in adult cancers, combined with compelling pediatric preclinical data, position PROTACs as a promising emerging therapeutic platform in pediatric oncology. Efforts to translate PROTACs into pediatric clinical trials are expected to advance as compounds mature and pediatric-specific targets are identified and validated (Table 2).

In addition to these strategies, recent advances have led to the development of new methods that enable the direct inhibition of oncogenic TFs. One example is OMO-103 (Omomyc mini-protein), a cell-penetrating mini-protein designed to inhibit MYC. Omomyc works by disrupting MYC ability to bind DNA and activate gene transcription. It forms dimers with MYC and MAX proteins that cannot attach to the E-box DNA sequences, where MYC acts to turn on target genes [201]. A Phase 2 pilot study is currently evaluating the safety and antitumor efficacy of intravenously administered OMO-103 in patients with advanced high-grade osteosarcoma (NCT06650514).

An alternative approach involves indirectly targeting fusion TFs. In Ewing sarcoma, for instance, inhibition of the EWS::FLI1 fusion protein can be achieved by blocking its epigenetic co-activators, P300 and CBP. Since EWS::FLI1 depends on these acetyltransferases to sustain enhancer activity and regulate tumor-promoting genes, inhibiting their enzymatic functions leads to EWS::FLI1 destabilization and enhanced therapeutic sensitivity. Notable examples include iCBP4 (derived from CCS1477), which selectively targets EP300/CBP in Group 3 MB, and the bromodomain inhibitor iP300w, which prevents P300/CBP-mediated histone acetylation, thereby disrupting EWS::FLI1-driven oncogenic signaling in Ewing sarcoma [202,203].

## 5. Concluding Remarks

It is now well established that the dysregulation of TFs, whether through direct genetic alterations or indirectly via disruptions in upstream signaling pathways, can lead to aberrant gene expression programs. These transcriptional changes play a pivotal role in driving tumor initiation, progression, and the acquisition of the hallmarks of cancer. Historically considered “undruggable”, TFs have emerged in recent years as promising therapeutic targets due to advances in molecular biology, drug development and delivery technologies. Novel strategies aimed at directly or indirectly modulating TF function are now demonstrating encouraging preclinical and clinical outcomes across various cancer types. Importantly, these approaches offer the potential for greater specificity and reduced toxicity compared to traditional chemotherapeutic agents. In parallel, an expanding body of research continues to uncover the multifaceted roles of TFs, extending beyond their canonical function as gene expression regulators. These insights include their involvement in epigenetic remodeling, chromatin architecture, cellular metabolism, immune evasion, and therapy resistance, further reinforcing the therapeutic value of targeting TFs in cancer. Despite this progress, the majority of clinical efforts remain focused on adult malignancies. There is a pressing need to extend these advancements to pediatric oncology, where the unique biology of childhood cancers requires specialized strategies. Increased investment in pediatric-specific research, preclinical modeling, and clinical trial infrastructure will be essential to bridge this gap. With continued innovation and a concerted push to address pediatric applications, targeting TFs may soon become a cornerstone of precision oncology across all age groups.

## Figures and Tables

**Figure 1 cancers-17-03439-f001:**
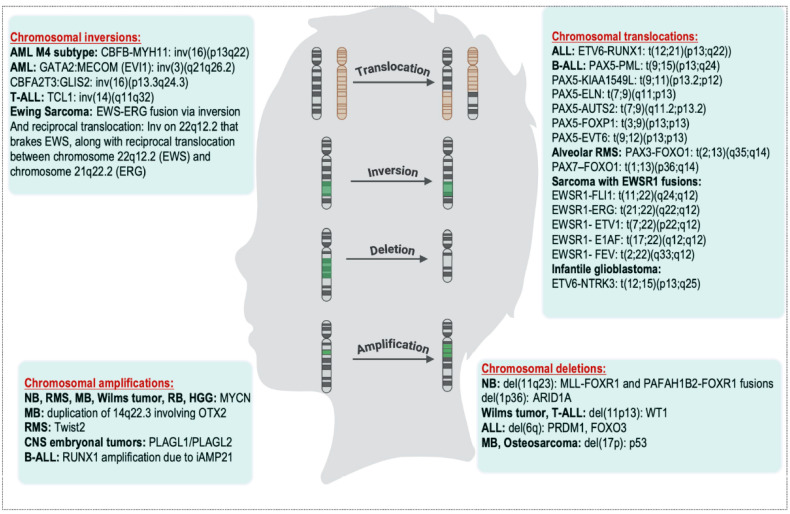
Key gene fusions and translocations identified in diverse pediatric tumor types.

**Figure 2 cancers-17-03439-f002:**
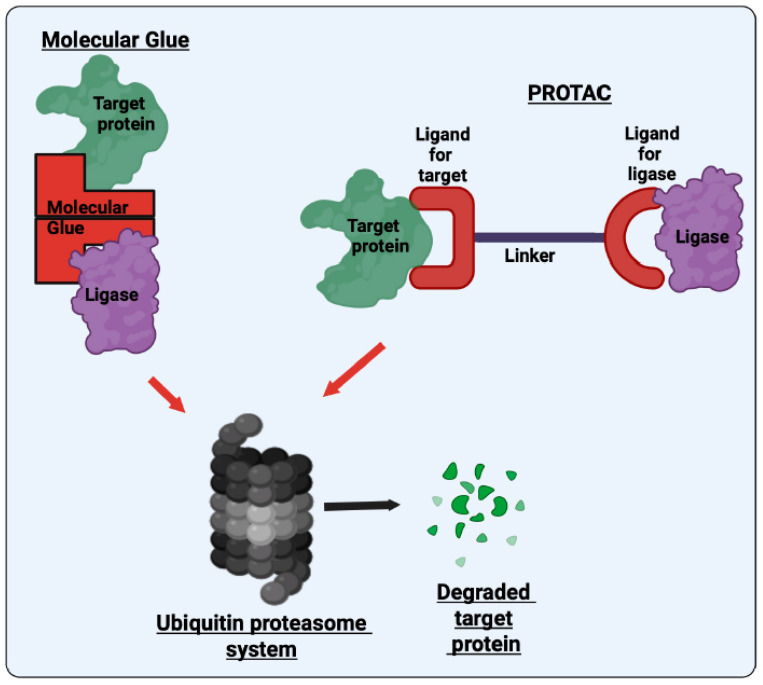
Basic mechanisms of molecular glue degraders and PROTACs. Molecular glues bring an E3 ubiquitin ligase and the target protein closer by stabilizing their interaction, which leads to ubiquitination and degradation. In contrast, PROTACs are bifunctional molecules that connect the E3 ligase and target protein through two separate binding domains, allowing for targeted ubiquitination and proteasomal degradation.

**Table 1 cancers-17-03439-t001:** Chromosomal aberrations and associated TFs in Pediatric Cancers.

Chromosomal Abnormality	Cancer Type	TFs Involved	Genes Rearrangement	Incidence	Functional Impact
**Chromosomal translocations** **:**					
*TCF3::PBX1* (t(1;19)(q23;p13)) [14,15,16]	B-ALL	*TCF3* *PBX1*	*TCF3* on Chr19p13 is translocated to *PBX1* gene on Chr1q23	~5% of B-ALL	Oncogenic fusion TF
*IGH::MYC* t(8;14)(q24;q32) [17]	BL	*MYC*	*MYC* gene translocated from Chr8q24 to the IGH locus on Chr14q32	~75% of BL	High MYC expression
*RUNX1::RUNX1T1* (t(8;21)(q22;q22)) [18,19,20]	AML	*RUNX1*	*RUNX1* on Chr21 is translocated to *RUNX1T1* gene on Chr8	~4–8% of AML	Fusion gene block RUNX1 functions
*PML::RARA* (t(15;17)(q24;q21)) [21]	APL	*PML* *RARA*	*PML* on Chr15 is fused with *RARA* gene on Chr17	>90% of APL	It acts as transcriptional repressor and blocks differentiation
*TCRD::LMO2* t(11;14)(p13;q11) [22,23]	T-ALL	*LMO2*	*LMO2* on Chr11p13 is translocated next to the regulatory elements of *TCRD* locus on Chr14q11	~7% of T-ALL	LMO2 overexpression via TCR enhancer hijacking
*NPM1::ALK* t(2;5)(p23;q35) [24]	ALCL	*NPM1*	*NPM1* gene on Chr5q35 is fused to the *ALK* gene on Chr2p23	~80–90% of ALCL	NPM1::ALK is constitutively active RTK
*EWSR1::WT1* t(11;22)(p13;q12) [25,26,27]	DSRCT	*WT1*	This reciprocal translocation fuses *EWSR1* gene on Chr22q12 with *WT1* on Chr11p13	100% cases	EWSR1::WT1 acts as a powerful chromatin activator
*ETV6::NTRK3* t(12;15)(p13;q25) [28,29,30]	CFS CMN	*ETV6*	Translocation fuses the N-terminal SAM domain of *ETV6* on Chr12p13 to C-terminal PTK domain of *NTRK3* on Chr15q25.	~70% of CFS	ETV6::NTRK3 is constitutively active RTK
*ASPSCR1::TFE3* t(X;17)(p11;q25) [31,32,33,34]	ASPS	*TFE3*	This non-reciprocal translocation fuses *ASPSCR1* on Chr17q25 with the *TFE3* gene on Chr Xp11	<1% of soft-tissue sarcomas	ASPSCR1::TFE3 is oncogenic TF
**Chromosomal inversions:**					
inv(7)(p15q34) [35,36]	T-ALL	*HOXA10* *HOXA11*	It places *HOXA* cluster genes near TCRβ locus, overexpressing *HOXA10* and *HOXA11*	rare	Inversion blocks T-cell differentiation
**Chromosomal deletions:**					
del(6q) q13-q25 [37,38,39]	ALL	*PRDM1* *FOXO3*	del(6q) leads to loss of tumor suppressor genes *PRDM1* and *FOXO3*	4–15%	Loss of PRDM1 and FOXO3 activity promotes leukemia
del(12p) [40]	ALL	*ETV6*	*ETV6* located at 12p13 is deleted due to del(12p). Deletion strongly associated with the t(12;21)(p13;q22)	~15% of ALL	Genes loss contribute to leukemia pathogenesis
del(7p or 7q) [41,42,43,44]	ALL AML	*IKZF1* *ZNFN1A1* *GLI3* *CUX1*	*IKZF1* on 7p12, *GLI3* on 7p13 and *CUX1* on 7q22 deleted in del(7p) or del(7q) leukemia	4–5%	IKZF1, ZNFN1A1 and CUX1 are tumor suppressors
del(5q) [45,46]	AML	*EGR1* *APC*	*EGR1* at 5q31.2 and *APC* at 5q22.2 are key genes haploinsufficient due to del(5q)	1.5%	EGR1 and APC are tumor suppressors
del(17p) [47,48,49]	SHH-MB	*TP53*	*TP53* located at 17p13.1. deleted due to del(17p). 17p deletion occurs together with gain or duplication of 17q	25–50%	p53 loss promote tumor progression
del(10q) [50,51]	MB	*MXI1*	Deleted 10q region harbors *MXI1* gene on chromosome 10q24 (around 10q24–q25 region)	15–20%	MXI1 is a transcriptional repressor
del(17p) [52,53]	Osteosarcoma	*TP53*	Deleted 17p region harbors *TP53* gene on 17p13.1.	40–70%	p53 loss promote tumor progression
del(1p36) [54,55]	Hepatoblastoma	*CAMTA1 CASZ1*	These TFs deleted due to 1p36 deletion	27%	CAMTA1 and CASZ1 are tumor suppressors
**Chromosomal amplifications:**					
1q Gain [56,57]	WT	*MYOG*	*MYOG* located at 1q and amplified due to 1q gain	~30%	MYOG linked to differentiation and chemotherapy resistance
iAMP21 [58,59,60]	B-ALL	*RUNX1*	Extra copies of the *RUNX1* gene on Chr21 due to complex rearrangements within the chromosome	2–5%	RUNX1 along with other gene amplifications in this region drive leukemia
iAmp-PAX5 [61,62,63]	B-ALL	*PAX5*	Extra copies of internal region of the *PAX5* gene are gained, rather than amplification of the entire gene	0.5–1.4%	PAX5 amplification disrupts B-cell differentiation
6p12-p21 [64,65,66]	Osteosarcoma	*RUNX2*	6p12-p21 amplification leads to increased *RUNX2* copy number	16–75%	RUNX2 acts as an oncogenic TF

Chr: Chromosome, TCF3: Transcription factor 3, PBX1: Pre-B-cell leukemia homeobox 1, IGH: Immunoglobulin heavy chain, BL: Burkitt lymphoma, PML: Promyelocytic leukemia, RARA: Retinoic acid receptor alpha, APL: Acute promyelocytic leukemia, TCRD: T-cell receptor delta TCRD, ALCL: ALK-positive anaplastic large-cell lymphoma, RTK: Receptor tyrosine kinase, EWSR1: Ewing sarcoma breakpoint region 1, DSRCT: Desmoplastic small round cell tumor, NTRK3: Neurotrophic tyrosine receptor kinase 3, CFS: Congenital (infantile) fibrosarcoma, CMN: Congenital mesoblastic nephroma (cellular subtype), ASPS: Alveolar Soft Part Sarcoma, EGR1: Early Growth Response 1, APC: Adenomatous Polyposis Coli, WT: Wilms Tumor, iAMP21: Intrachromosomal amplification of chromosome 21, iAMP: intragenic amplification.

**Table 2 cancers-17-03439-t002:** Therapeutic protein degraders targeting TFs in pediatric cancers.

Name	E3 Ubiquitin Ligase	Target TFs	Cancer Type	Clinical Status
**Molecular Glue Degraders:**				
Lenalidomide [176,177]	CRBN	IKZF1, IKZF3	Low-grade gliomas: pilocytic astrocytoma, optic pathway glioma, relapsed or refractory AML	Phase II
Pomalidomide [178,179]	CRBN	IKZF1, IKZF3	Recurrent, progressive/refractory CNS tumors	Phase I
Thalidomide alone (NCT03257631) or in combination with cyclophosphamide (NCT00003754), chemo (NCT06470464), carboplatin (NCT00179881)	CRBN	IKZF1, IKZF3, SALL4	Recurrent or progressive primary brain tumors, recurrent or refractory childhood cancers, yolk sac tumor and pediatric brain stem gliomas.	Phase II
**PROTACs:**				
ARV-825 [180,180,181]	CRBN	BRD2, BRD3, BRD4, MYC	T-ALL Neuroblastoma	Preclinical
MZ1 [182,183]	VHL	BRD2, BRD3, BRD4, MYC, MYCN	AML Neuroblastoma	Preclinical
dBET1 [184]	CRBN	BRD2, BRD3, BRD4	AML	Preclinical
MS40 [185]	CRBN	IKZF1, IKZF3	MLL-rearranged leukemias	Preclinical
PROTAC 8b [186]	VHL	BRD4	BAL	Preclinical
GNE-987 [187,188]	VHL	BRD4	Osteosarcoma AML	Preclinical
**PROTACs targeting transcriptional regulators (TRs):**				
**Name**	**E3 Ubiquitin Ligase**	**Target TRs**	**Cancer Type**	**Status**
ACBI-1 [189]	VHL	SMARCA2 SMARCA4 (chromatin remodeler)	Alveolar RMS	Preclinical
dTRIM24 [190]	VHL	TRIM24 (transcriptional activator)	AML	Preclinical
AU-15330 [191]	VHL	SMARCA2 SMARCA4	DIPGs	Preclinical
753B [192]	VHL	BCL-XL BCL-2 (anti-apoptotic)	AML	Preclinical
JQAD1 [193]	CRBN	EP300	Neuroblastoma	Preclinical
CP-10 [194]	CRBN	CDK6	AML	Preclinical
YX-2-107 [195]	CRBN	CDK6	Ph^+^ALL	Preclinical

Biphenotypic acute leukemia (BAL).

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
