# Peer review of "Masters of Gene Expression: Transcription Factors in Pediatric Cancers"

_cancers, 2025, doi:10.3390/cancers17213439_

Round 1
Reviewer 1 Report
Comments and Suggestions for Authors
This comprehensive review provides valuable coverage of transcription factor alterations in pediatric cancers and therapeutic targeting approaches. The manuscript would benefit from minor revisions to enhance its currency and accuracy:
- Condense introduction for improved focus and readability.
- Incorporate recent ETV6-RUNX1 breakthroughs. The review lacks critical updates on molecular and pharmacological heterogeneity of ETV6::RUNX1 acute lymphoblastic leukemia and genomic determinants of therapy response (Li et al., https://doi.org/10.1038/s41467-025-56229-7
Oksa et al., https://doi.org/10.1038/s41375-025-02683-7 ) - Include complex structural variation findings. Recent work demonstrates that complex structural variations creating "rearrangement bursts" are more prevalent in pediatric solid tumors than previously recognized (van Belzen et al., Cell Genom 4:100675, 2024). This context would complement the current focus on individual translocations.
- Correct Table 1: RUNX1-RUNX1T1 t(8;21) should be listed under AML rather than ALL, with appropriate frequency adjustments to match established clinical knowledge.
- Consider direct TF targeting advances: Brief mention of OMOMYC mini-protein in Phase 2 pediatric trials (NCT06650514) and indirect EWS-FLI1 targeting through P300/CBP inhibition would update the therapeutic targeting section.
- Typographical errors and nomenclature in Table 1: “PAX5-EVT6” should be PAX5-ETV6; “KZF1” should be IKZF1. Gene and fusion names should be standardized according to HGNC guidelines.
Author Response
This comprehensive review provides valuable coverage of transcription factor alterations in pediatric cancers and therapeutic targeting approaches. The manuscript would benefit from minor revisions to enhance its currency and accuracy:
- Condense introduction for improved focus and readability.
Reply: I sincerely thank the reviewer for their constructive feedback and valuable suggestions that helped improve the quality of the manuscript. Following the recommendation, I condensed the introduction to enhance its focus and improve overall readability.
Incorporate recent ETV6-RUNX1 breakthroughs. The review lacks critical updates on molecular and pharmacological heterogeneity of ETV6::RUNX1 acute lymphoblastic leukemia and genomic determinants of therapy response (Li et al., https://doi.org/10.1038/s41467-025-56229-7
Oksa et al., https://doi.org/10.1038/s41375-025-02683-7 )
Reply: As suggested by the reviewer, recent breakthroughs related to ETV6::RUNX1 reported in the following studies have been incorporated and discussed in the revised manuscript. (Section 2.1 Chromosomal translocations)
Include complex structural variation findings. Recent work demonstrates that complex structural variations creating "rearrangement bursts" are more prevalent in pediatric solid tumors than previously recognized (van Belzen et al., Cell Genom 4:100675, 2024). This context would complement the current focus on individual translocations.
Reply: Findings related to complex structural variations have been included and discussed in the revised manuscript. (Section 2.5: Complex Structural Variations)
Correct Table 1: RUNX1-RUNX1T1 t(8;21) should be listed under AML rather than ALL, with appropriate frequency adjustments to match established clinical knowledge.
Reply: Table 1 has been updated to list RUNX1-RUNX1T1 t(8;21) under AML, with the frequency adjusted accordingly.
Consider direct TF targeting advances: Brief mention of OMOMYC mini-protein in Phase 2 pediatric trials (NCT06650514) and indirect EWS-FLI1 targeting through P300/CBP inhibition would update the therapeutic targeting section.
Reply: As suggested by the reviewer, I have updated the therapeutic targeting section to include the OMOMYC mini-protein in Phase 2 pediatric trials (NCT06650514) and indirect EWS-FLI1 targeting via P300/CBP inhibition. (Section 4)
Typographical errors and nomenclature in Table 1: “PAX5-EVT6” should be PAX5-ETV6; “KZF1” should be IKZF1. Gene and fusion names should be standardized according to HGNC guidelines.
Reply: The typographical and nomenclature errors in Table 1 have been corrected. Gene and fusion names have been standardized according to HGNC guidelines.
Reviewer 2 Report
Comments and Suggestions for Authors
1.MyoD1 mutation is usually in spindle cell rhabdomyosarcoma not embryonal RMS.
2. Gene alterations also include point mutations such as CTNNB1 and gain ,you can also add some paragraph about these kinds of alterations.
3.MYCN amplification can also appear in a subset of alveolar rhadomyosarcoma.You can add some sentences to the MYCN amplification paragraph section.
Author Response
MyoD1 mutation is usually in spindle cell rhabdomyosarcoma, not embryonal RMS.
Reply: I sincerely appreciate the reviewer’s constructive feedback and insightful suggestions, which have significantly improved the quality of the manuscript. The text has been corrected accordingly
- Gene alterations also include point mutations such asCTNNB1 and gain ,you can also add some paragraph about these kinds of alterations.
Reply: As suggested by the reviewer, point mutations such as CTNNB1 and gene gain events have been included and discussed in Section 3 of the revised manuscript.
3.MYCN amplification can also appear in a subset of alveolar rhadomyosarcoma. You can add some sentences to the MYCN amplification paragraph section.
Reply: In the revised manuscript, I have included a discussion of MYCN amplifications in a subset of alveolar rhabdomyosarcoma. (Section 2.4)